# Determination of Free Fatty Acids in Krill Oil during Storage Based on NH_2_-MMS

**DOI:** 10.3390/foods13172736

**Published:** 2024-08-28

**Authors:** Shibing Zhang, Yiran Wang, Chunyu Yang, Xi Wang, Siyi Wang, Jiping Yin, Yinan Du, Di Wu, Jiangning Hu, Qi Zhao

**Affiliations:** 1State Key Laboratory of Marine Food Processing and Safety Control, Dalian Polytechnic University, Dalian 116034, China; 13470306550@163.com (S.Z.); wyr990112@163.com (Y.W.); mengnvhaiqiqi@163.com (C.Y.); wangxi9215@icloud.com (X.W.); www011478@163.com (S.W.); dyn7381@163.com (Y.D.); m13039998695@163.com (D.W.); hujiangning2005@hotmail.com (J.H.); 2National Engineering Research Center of Seafood, School of Food Science and Technology, Dalian Polytechnic University, Dalian 116034, China; 3Information Technology Center, Dalian Polytechnic University, Dalian 116034, China; yinjp@dlpu.edu.cn

**Keywords:** ZSM-5, krill oil, solid-phase extraction, GC-FID

## Abstract

In this study, amino-modified micro-mesoporous silica (NH_2_-MMS) with hierarchical pores was prepared by modifying micro-mesoporous silica ZSM-5 with 3-aminopropyltriethoxysilane and used as an adsorbent in solid-phase extraction to analyze free fatty acids (FFAs) in krill oil during storage for an initial time. The Brunner Emmet Teller adsorption experiment and Fourier transform infrared spectroscopy demonstrate that NH_2_-MMS, with a hierarchical pore structure, was successfully synthesized. The adsorption experiments, especially static adsorption, indicate that the absorption ability of the prepared NH_2_-MMS, with a hierarchical pore structure, toward FFAs was better than that of traditional amino-modified mesoporous silica (SBA-15) with a mesoporous structure at all temperature and concentrations. Fairly low limits of detection (0.06–0.15 μg g^−1^), acceptable recoveries (85.16–94.31%), and precision (0.08–5.26%) were attained under ideal circumstances. Moreover, NH_2_-MMS has the advantages of easy preparation and being environmentally friendly. As a result, this method offers an alternative to the current method for determining FFAs in different kinds of oil specimens.

## 1. Introduction

As one of the important marine crust ocean organisms living in the Antarctic Ocean, Antarctic krill (*Euphausia superba*) have attracted increasing interest worldwide for their huge biomass abundance and rich nutritional value [1,2]. Antarctic krill are usually processed into krill meal and used as bait in aquaculture industries [3]. In order to widen the utilization possibilities of Antarctic krill resources, Antarctic krill oil (AKO), with its abundant phospholipids (PLs) and bound ω-3 polyunsaturated fatty acids (PUFAs), has always been developed as a good source for the production of functional oils and regarded as a vital high-priced ingredient of Antarctic krill [4,5].

Being rich in PLs and PUFAs, the oxidation and rancidity of lipids in AKO are susceptible to occur [6]. As one important class of lipids, free fatty acids (FFAs) are more prone to oxidation than triglycerides and PLs [7]. Due to these unstable chemical characteristics, the level of FFA content in AKO directly corresponds to the quality of the oil products [8]. Hence, considering the crucial function of FFA content in the evaluation of oil quality, there is clearly a demand to discover a low-priced and trustworthy analytical technique to trace and determine the presence of FFAs in AKO.

Usually, FFA analysis of AKO is carried out by titrating with an alkaline solution until the phenolphthalein endpoint, which is tedious, labor-intensive, and solvent-consuming [9]. To overcome these issues, several methods based on the colorimetric spectrum [10], near-infrared spectroscopy (NIR) [11], nuclear magnetic resonance spectroscopy (NMR) [12], gas chromatography (GC) [13], or high-performance liquid chromatography (HPLC) [14] have been developed. Although the use of NIR and NMR is convenient and rapid, it presents some problems, such as low sensitivity and overlapping signal peaks, and their application is limited to a certain extent. In comparison, GC and HPLC are widely employed techniques for both the qualitative and quantitative assessment of FFAs. Consequently, the combination of GC and a flame ionization detector (FID) is better for analyzing FFAs.

With the aim of avoiding the influence caused by glycerides as well as PLs prior to the derivatization step, solid-phase extraction (SPE) is widely employed to separate and concentrate FFAs from oil specimens. Hence, the choice of adsorbent is of great importance to remove interference and improve recoveries. To date, various adsorbents, including C_18_ [15], silica [16], polydimethylsiloxane [17], and amino-modified silica [18], have been used as adsorbents. Of special interest is amino-modified silica because of the special interaction between aminopropyl and FFAs. However, the application of traditional silica materials is limited because of their low adsorption capacity.

To settle the above shortcomings, numerous silica adsorbents with high specific surface areas, such as microporous (<2 nm), mesoporous (2–50 nm), and macroporous (>50 nm), as well as mesoporous materials with hierarchical pores [19,20], were developed. Among them, mesoporous materials with hierarchical pores, including micro-mesoporous, macro-mesoporous, and macro–meso-microporous structures, are of great potential to improve adsorption and separation behaviors based on their unique porous structures and larger surface areas [19,21,22]. Owing to the easier accessibility of pollutants in interconnected mesopores and the larger surface area of microporous structures, hierarchical porous silica with a micro-mesoporous structure is ideal as an adsorbent in SPE [23,24]. Up to now, the studies of micro-mesoporous silica mainly focus on the synthesis method and mechanism [22,25], but there is little research on how to apply them for adsorption and analysis application, especially in the extraction and separation of analytes from oil specimens.

Among micro-mesoporous silica, hierarchical ZSM-5, especially micro-mesoporous ZSM-5, which can be produced from a variety of silicon (Si) and aluminum (Al) sources, has attracted widespread attention due to its unique MFI channel structure, high acid strength, and thermal stability [26,27]. Inspired by this, 3-aminopropyltriethoxysilane (APTES) was modified to produce ammoniated micro-mesoporous ZSM-5 (NH_2_-MMS). Scanning electron microscopy (SEM), transmission electron microscopy (TEM), energy spectrum analysis (EDS), X-ray diffraction (XRD), Fourier transform infrared spectroscopy (FTIR), and nitrogen adsorption/desorption studies were used to thoroughly characterize the resultant NH_2_-MMS. Then, using both static and dynamic adsorption tests, the adsorption capacities of NH_2_-MMS and ammoniated mesoporous silica (NH_2_-SBA-15) were contrasted. Following the selection of the optimum extraction conditions, NH_2_-MMS was used as an SPE adsorbent to analyze the FFA content in AKO during storage.

## 2. Experiments

### 2.1. Chemicals and Materials

Analytical grades of methanol, n-hexane, toluene, APTES, and dichloromethane were purchased from Aladdin Reagent Co. (Shanghai, China). ZSM-5 (n(SiO_2_):n(Al_2_O_3_) = 40) and SBA-15 (pore size: 6.0–11.0 nm, specific surface area: 600.0–800.0 m^2^) were purchased from Sigma–Aldrich Co. (Shanghai, China). Standards of myristic acid (C14:0), palmitic acid (C16:0), stearic acid (C18:0), oleic acid (C18:1), linoleic acid (C18:2), eicosanoic acid (C20:1n-9), and margaric acid (C17:0) were purchased from TCL (Shanghai, China).

The AKO storage experiment served as the source of the information collected from the Liaoyu Group Corporation (Dalian, China) and was carried out as follows: after being randomly divided into four groups, each AKO sample (>250 mL) was packed in a slightly closed brown-lid glass bottle and separately stored at −20 °C, 4 °C, 25 °C, and 40 °C for 35 d. Then, the samples (>50 mL) were randomly collected at 7d intervals and kept at −80 °C until the unified analysis.

### 2.2. The Preparation of NH_2_-MMS

First, 0.2 g of obtained ZSM-5 was dissolved with some agitation in 100 mL of anhydrous toluene. Next, the mixture was combined with 0.2 mL of APTES. After 24 h of agitation at 80 °C, the resultant precipitate was centrifuged and washed repeatedly with anhydrous ethanol to obtain NH_2_-MMS. For comparison of the adsorption properties, the corresponding NH_2_-SBA-15 was prepared in the same manner as NH_2_-MMS, except ZSM-5 was substituted by SBA-15.

### 2.3. The Characterization of NH_2_-MMS

TEM (JEM-2100, UHR, Tokyo, Japan) and SEM (JSM-7800F, JEOL, Tokyo, Japan) were utilized to examine the morphology of NH_2_-MMS. The chemical and elemental composition was analyzed by FTIR spectrometry (Spectrum10, PerkinElmer, Waltham, MA, USA), XRD (XRD-6100, Shimadzu, Tokyo, Japan), and EDS (X-Max50, JEOL, Hertfordshire, UK). Using a Quadrasorb surface area analyzer (SI-MP-21; Quantachrome, Graz, Austria), BET nitrogen adsorption–desorption tests were used to determine the pore distribution.

### 2.4. Adsorption Experiments

The static adsorption experiment was executed as follows: firstly, 2.0 mL of standard solutions of palmitic acid (C16:0) containing concentrations of 0.5, 1.0, 1.5, 2.0, 5.0, 6.0, 8.0, 9.0, and 10 mmol L^−1^ were added to 20.0 mg of NH_2_-MMS. The mixture was filtered after standing at 35 °C for 24 h. Subsequently, the filtrate underwent alteration through the derivatization step and was assessed using GC–FID. The sorption of NH_2_-MMS at 45 °C and 55 °C was investigated in the same manner as at 35 °C. Furthermore, the static sorption experiment of NH_2_-SBA-15 was identical to the previous one, except the absorbent was replaced by NH_2_- SBA-15.

In this experiment, the adsorption process of NH_2_-MMS and NH_2_-SBA-15 on palmitic acid (C16:0) was simulated using the Langmuir and Freundlich isothermal adsorption equations [28].

The Langmuir isotherm model’s Equation (1) is as follows:(1)$$\frac{1}{q_{e}}=\frac{1}{q_{m}}+\frac{1}{q_{m}c_{e}K_{L}}$$

The Freundlich isotherm model’s Equation (2) is as follows:(2)$${\mathrm{lg}q}_{e}=\mathrm{lg}K_{F}+\frac{1}{n}\mathrm{lg}c_{e}$$
where the concentration and adsorption capacity of palmitic acid (C16:0) at equilibrium are represented by *q_e_* and *c_e_*, respectively; *q_m_* is the maximal saturation adsorption capacity; *K_L_* is the Langmuir constant; *K_F_* is the Freundlich constant; and *n* is the equation constant.

This is how the dynamic adsorption experiment was executed: after 20 mg of NH_2_-MMS was concentrated in a standard solution of 0.5 mmol L^−1^ palmitic acid (C16:0), the resulting solution was kept at 25 °C. At 20, 30, 40, 60, 120, 240, 300, 360, 420, 480, and 1440 min, samples were taken, and the samples were processed and examined in the same manner as previously described. The dynamic sorption experiment for NH_2_-SBA-15 was carried out similarly to the previous one, with the exception that the absorbent was switched.

To greater comprehend the adsorption kinetics of NH_2_-MMS and NH_2_-SBA-15 on palmitic acid (C16:0), first-order kinetic and second-order kinetic adsorption models [28] were utilized in this experiment to simulate the dynamic adsorption. The equations of the first-order dynamic model (3) and the second-order dynamic model (4) are as follows:(3)$$\mathrm{lg}\left(q_{e}\right.-\left.q_{t}\right)=\mathrm{lg}q_{e}-\frac{K_{1}t}{2.303}$$
(4)$$\frac{t}{q_{t}}=\frac{1}{K_{2}q_{e}^{2}}+\frac{t}{q_{e}}$$
where the adsorption capacity of palmitic acid (C16:0) at equilibrium is represented by *q_e_*; *q_t_* is the adsorption capacity of palmitic acid (C16:0) at adsorption time *t*; *K*_1_ is the first-order rate constant; and *K*_2_ is the second-order rate constant. 

### 2.5. SPE Procedure

First, 120.0 mg of NH_2_-MMS was compressed into an empty polytetrafluoroethylene column to make a self-prepared column. Following activation with n-hexane (4.0 mL), the column was loaded with 1.0 mL of krill oil specimens (10 mg mL^−1^), with 0.2 mg of heptadecanoic acid (C17:0) as an internal reference. After flushing with 4.0 mL of n-hexane/ethyl ether (85:15, *v*/*v*) and 4.0 mL of n-hexane/ethyl acetate (15:85, *v*/*v*) in order, 4.0 mL of a mixed solution of acetic acid and ethyl ether (2:98, *v*/*v*) was used to elute FFAs, and the eluent was collected.

### 2.6. Derivatization

After the eluent was dried under protection with nitrogen, 2.0 mL of a 0.5 mol L^−1^ sodium hydroxide methanol solution was added. Refluxing the mixture at 80 °C for five minutes, 2.0 mL of boron trifluoride methanol solution was added. After bringing the combined solution down to room temperature, 1.5 mL of hexane was added for extraction. After that, 0.5–1.0 g of anhydrous sodium sulfate was added, and the upper hexane extract was collected in an EP tube. After 1 min of vortexing, the mixture was allowed to rest for 1–2 h. Finally, the upper solution was collected and then fixed to 1.0 mL before being determined by GC-FID.

### 2.7. GC-FID Analysis

To analyze and isolate the obtained fatty acid methyl esters (FAMEs), GC-FID (Agilent 7890B; Agilent Technologies, Santa Clara, CA, USA) was used in conjunction with a fused silica capillary column (SP-2560; 100 m × 0.25 mm × 0.20 μm; Supelco, Bellefonte, PA, USA). The programmed oven temperature was 120 °C for 9 min; 120–200 °C at 20 °C per minute and 200 °C for 5 min; and 200–230 °C at 30 °C per minute and 230 °C for 10 min. The instrument’s injector and detector reached temperatures of 220 and 260 °C, respectively. Furthermore, a split ratio of 100:1 was used with an injection volume of 1.0 μL. In continuous pressure mode, the flow rate through the transport gas (N_2_) was 2.0 mL min^−1^. The qualitative analyses of FAMEs were carried out according to the retention time of each FFA, and the quantitative analyses of FAMEs were carried out according to the combination of external and internal methods. Hence, the standard curves for each FFA were plotted using the concentration of each FFA serving as the horizontal axis and the peak area ratio of each FFA to IS serving as the vertical axis.

### 2.8. Statistical Analysis

Each experiment had three duplicates, and the values are given as mean ± standard deviation (SD). The statistical analysis was utilized to conduct the SPSS 24.0 program (SPSS Inc., Chicago, IL, USA). Variations between the indicators were subjected to univariate analysis of variance (Student–Newman–Keuls post hoc test), with *p* < 0.05 deemed to be statistically noteworthy.

## 3. Results and Discussion

### 3.1. Characterization of NH_2_-MMS

The morphology of ZSM-5 and NH_2_-MMS was characterized by SEM and TEM (Figure 1). The SEM of ZSM-5 (Figure 1A) and NH_2_-MMS (Figure 1B) showed a loose porous structure, which means that the pore structure of ZSM-5 did not change during amination. Parallel pore structures can be observed (marked in red) in the TEM of ZSM-5 (Figure 1C) and NH_2_-MMS (Figure 1D), indicating that ZSM-5 and NH_2_-MMS have mesoporous structures and were unchanged after modification [29,30], which is consistent with the SEM results. The even distribution of Si, C, Al, and N elements in the elemental distributions of ZSM-5 (Figure 2A) and NH_2_-MMS (Figure 2B,C) indicate good dispersion in the prepared NH_2_-MMS. The elemental composition of EDS in ZSM-5 and NH_2_-MMS is shown in Table 1. In addition, the presence of the N element in the EDS of NH_2_-MMS proves that APTES was successfully modified on the surface of ZSM-5.

The FTIR spectra for ZSM-5 and NH_2_-MMS are shown in Figure 2D. The shoulder peak at 1230, 807, and 1109 cm^−1^ was the asymmetric telescopic vibration, external symmetric stretching vibration, and internal symmetric vibration peak of Si-O-Si, respectively [31]. Moreover, the strong peaks at 550 cm^−1^ of ZSM-5 and NH_2_-MMS were the characteristic peaks of MFI topological molecular sieves, which further indicate that the original structure of ZSM-5 was not broken after the graft of the amino group [32]. However, the distinctive absorption peaks of NH_2_-MMS at 2927 cm^−1^ and 2857 cm^−1^ (the stretching vibration of -C-H) and 1470 cm^−1^ (the stretching vibration peak of N-H) [31] further suggest that amino groups were successfully grafted to the surface of ZSM-5.

The crystal structures of ZSM-5 and NH_2_-MMS were ascertained by XRD. As shown in Figure 2E, peaks appeared at 2θ = 7.9°, 8.9°, 23.2°, 23.9°, and 24.4° and were the characteristic diffraction for the crystal plane of (101), (020), (501), (151) and (303), respectively, which represented the existence of an MFI topological molecular sieve structure in ZSM-5 and NH_2_-MMS [33]. In addition, no significant broadening of peak strength at 2θ = 7.9° and 8.9° was found after the amino group was grafted onto the surface of ZSM-5, indicating the crystal structure was not broken after amino grafting, which is consistent with the results of TEM.

The porous structure of ZSM-5 and NH_2_-MMS was determined by N_2_ adsorption–desorption experiments (Figure 2F). Type I isotherms were displayed in the adsorption–desorption isotherms when the relative pressure interval was 0.0 < P/P0 < 0.2, showing the presence of micropore structures in ZSM-5 and NH_2_-MMS [34]. The adsorption isotherms of typical IV isotherms with hysteresis loops were shown to rise with relative pressure P/P0, suggesting the presence of mesoporous structures in NH_2_-MMS and ZSM-5 [35]. The I + IV adsorption–desorption isotherms proved the hierarchical porous structures of ZSM-5 and NH_2_-MMS, and the hierarchical porous structures were not broken after amino grafting.

Based on N_2_ adsorption–desorption isotherms, the BET surface area, pore diameters, and total pore volumes of ZSM-5 and NH_2_-MMS are listed in Table 2. The pore diameters of ZSM-5 and NH_2_-MMS were 10.31 and 12.93 nm, respectively, and further indicated the existence of mesoporous structures in ZSM-5 and NH_2_-MMS. The increase in the pore diameter may have been induced by the reaming action of the amine and toluene solution.

### 3.2. The Adsorption Experiment of NH_2_-MMS

As one of the high-content fatty acids found in edible oil, palmitic acid (C16:0) was selected as the analyte to investigate the static adsorption of NH_2_-MMS toward FFAs at 35, 45, and 55 °C. The adsorbed amount of palmitic acid (C16:0) on NH_2_-MMS increased in a temperature-dependent way, as seen in Figure 3A, suggesting that the entire adsorption reaction was an endothermic process and increasing the temperature was advantageous for adsorption [36]. The adsorption capacities of NH_2_-MMS toward FFAs were further examined by computational modeling with Freundlich and Langmuir adsorption models. The correlation coefficients for the adsorption equations are shown in Table 3. The outcomes demonstrate that the adsorption of NH_2_-MMS toward FFAs was better described by the Freundlich adsorption equation (R^2^ = 0.991 > 0.957; 0.995 > 0.963; and 0.998 > 0.916). The values of parameter n were all over 1.0, suggesting that the adsorption process of NH_2_-MMS toward FFAs was favorable [37].

In order to evaluate the effect of hierarchical pores on adsorption, the adsorbed amount of palmitic acid (C16:0) on single mesoporous NH_2_-SBA-15 at different temperatures was carried out (Table 3). The adsorption of NH_2_-MMS and NH_2_-SBA-15 toward palmitic acid (C16:0) at 55 °C is shown in Figure 3B. The adsorbed amount of palmitic acid (C16:0) on hierarchical pore NH_2_-MMS was clearly higher than that of single mesoporous NH_2_-SBA-15 at all concentrations, and the improvement in NH_2_-MMS toward NH_2_-SBA-15 may be due to the existence of a multistage pore structure in NH_2_-MMS. In addition, the adsorption ability of the absorbent can be reflected in the value of K_F_. The constants K_F_ for NH_2_-MMS and NH_2_-SBA-15 were 0.175 and 0.106 at 55 °C, respectively. This result is consistent with the adsorbed amount in Figure 3B.

The adsorption capacities of NH_2_-MMS and NH_2_-SBA-15 toward palmitic acid (C16:0) rose with time, as demonstrated in Figure 3C. According to the results of the static adsorption experiment, there is a certain improvement in the adsorption concentration of palmitic acid on NH_2_-MMS at the adsorption equilibrium compared to NH_2_-SBA-15. This result is consistent with the data obtained by the static adsorption experiment.

Then, the adsorption dynamic behaviors of FFAs on NH_2_-MMS and NH_2_-SBA-15 were investigated using first- and second-order adsorption kinetics, and the relevant parameters for fitting are displayed in Table 4. The outcomes demonstrate that the quasi-second-order kinetic equation was more appropriate for NH_2_-MMS adsorption of palmitic acid (C16:0), as the R^2^ for the fitting curve of the second-order kinetic equation was significantly higher than that of the first-order kinetic equation. The above results suggest that the adsorption of FFAs on NH_2_-MMS was chemical adsorption and had two adsorption sites, probably an acid-base interaction and a hydrogen bond [38,39].

### 3.3. Optimization of Solid-Phase Extraction Conditions

To prevent TAG, DAG, PL, and MAG interference in oil specimens, 4.0 mL of n-hexane–ethyl ether (85:15, *v*/*v*) and 4.0 mL of n-hexane–ethyl acetate (15:85, *v*/*v*) were used for leaching. Then, the FFAs were eluted using 4.0 mL of acetic acid–ether (2:98, *v*/*v*) as the elution reagent. NH_2_-MMS, as the adsorbent, plays a significant role in the separation of distinct lipid classes, so the effect of the NH_2_-MMS amount on the FFA content in krill oil was evaluated, and the results are shown in Figure 4. As the amount of NH_2_-MMS concentration rose from 30.0 mg to 120.0 mg, the detected concentration of FFAs increased from 0.21–1.74 μg g^−1^ to 0.70–3.33 μg g^−1^. Then, the fatty acid content reached its maximum and showed no significant increases when the adsorbent amount further increased. Therefore, 120.0 mg of NH_2_-MMS was selected for subsequent experiments.

### 3.4. Linearity Range, LOQs, and LODs

Before the standard curves were established, the composition of FFAs in AKO was evaluated. Only myristic acid (C14:0), palmitic acid (C16:0), stearic acid (C18:0), oleic acid (C18:1), linoleic acid (C18:2), and eicosanoic acid (C20:1n-9) were detected, and the other PUFAs, including stearidonic acid (C18:4n-3), EPA, and DHA, as well as low-carbon chain fatty acids, were prone to oxidation and not detected because of low abundance [18,26]. Under the aforementioned ideal conditions, the standard curves for each FFA were plotted using the concentration of each FFA serving as the horizontal axis and the peak area ratio of each FFA to IS serving as the vertical axis. The correlation curves are shown in Figure 5, and the correlation coefficients, linear ranges, limits of quantification (LOQs), and limits of detection (LODs) are listed in Table 5. The findings demonstrate that good linearities in the range of 0.10–100 μg g^−1^, with R^2^ ranging from 0.992 to 0.999, were obtained. LODs (signal/noise ratio 3:1) and LOQs (signal/noise ratio 10:1) were in the range of 0.06–0.15 μg g^−1^ and 0.20–0.50 μg g^−1^, respectively.

### 3.5. Recoveries and Precision

The practicability of the method was gauged through the intraday and interday precision in krill oil added with 0.5, 10, and 50 μg g^−1^ of FFAs, including myristic acid (C14:0), palmitic acid (C16:0), palmitoleic acid (C16:1), stearic acid (C18:0), oleic acid (C18:1n-9), and arachidic acid (C20:0). As presented in Table 6, the relative standard deviations (RSDs%) of intraday and interday precision (1.10–5.12%) were less than 15%, and the recoveries (85.16–94.31) were in the range of 75% to 120%. These results indicate that our method meets the requirements of the AOAC guidelines [40] and can be suitable for the analysis of FFAs in krill oil specimens.

### 3.6. Applications in Real Samples

By analyzing the FFA content in AKO under various temperature conditions during storage, the established method was utilized to further demonstrate the viability of the suggested approach. The initial contents of myristic acid (C14:0), palmitic acid (C16:0), palmitoleic acid (C16:1), stearic acid (C18:0), oleic acid (C18:1n-9), and eicosanoic acid (C20:1n-9) were 0.00, 2.87, 0.00, 1.00, 0.79, and 1.62 μg g^−1^, respectively. After storage at −20 °C for 35d (Table 7), the corresponding values were increased to 0.68, 3.42, 0.83, 1.69, 1.02, and 2.46 μg g^−1^ and increased by 0.68, 0.55, 0.83, 0.69, 0.23, and 0.84 μg g^−1^, respectively. After storage at 4 °C for 35d (Table 8), the FFA contents increased to 0.98, 3.48, 0.78, 1.76, 1.51, and 2.57 μg g^−1^, respectively, and increased by 0.98, 0.78, 0.78, 0.76, 0.73, and 0.95 μg g^−1^, respectively. After storage at 25 °C for 35d (Table 9), the contents of these fatty acids increased to 1.27, 3.45, 1.00, 1.56, 2.38, and 2.88 μg g^−1^ and increased by 1.27, 0.75, 1.00, 0.56, 1.60, and 1.26 μg g^−1^, respectively. After storage at 40 °C for 35d (Table 10), the FFA contents increased to 1.58, 4.29, 1.46, 2.76, 2.87, and 4.18 μg g^−1^ and increased by 1.58, 1.60, 1.46, 1.76, 2.08 and 2.56 μg g^−1^, respectively. It can be seen that as the temperature rose, the FFA level of the krill oil increased dramatically, suggesting that the temperature was favorable for the breakdown of lipids in the oil. The results were with the provided by Fu et al. [41].

### 3.7. Comparison to the Existing Method

The generated adsorbents’ morphological characteristics, pore structure, amination reaction conditions, and loading capacity were compared to the matching adsorbent in the literature; the outcomes are displayed in Table 11. The produced NH_2_-MMS with hierarchical pores clearly offer the benefits of high adsorption capacity, ease of manufacture, and environmental friendliness over the current adsorbent utilized in FFA analysis. It could be regarded that this strategy offers a viable and different way to analyze FFAs in the future.

## 4. Conclusions

Our investigation successfully shows that NH_2_-MMS with hierarchical pores can effectively extract FFAs from krill oil specimens. The BET adsorption experiment and FTIR spectroscopy indicate that amino-modified ZMS-5 with a hierarchical pore structure was successfully synthesized. The self-prepared NH_2_-MMS adsorbent in our method has the advantage of being less expensive and ecologically friendlier than the existing adsorbent. This strategy is thought to offer a potentially different way to assess FFAs in order to ensure oil quality and cleanliness in the future.

## Figures and Tables

**Figure 1 foods-13-02736-f001:**
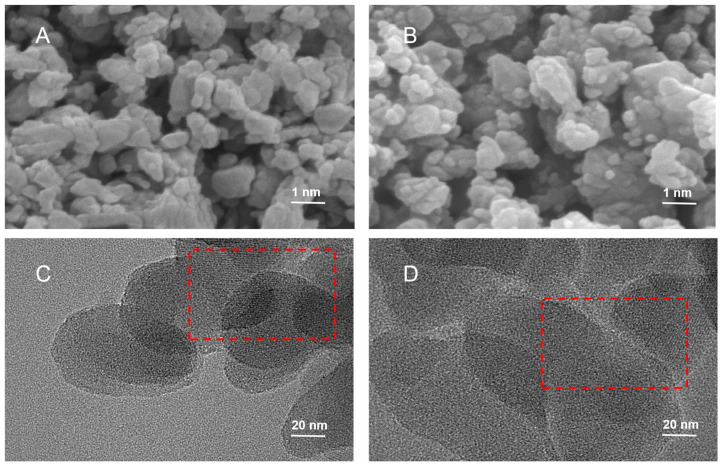
The SEM images of ZSM-5 (**A**) and NH_2_-MMS (**B**); the TEM of ZSM-5 (**C**) and NH_2_-MMS (**D**).

**Figure 2 foods-13-02736-f002:**
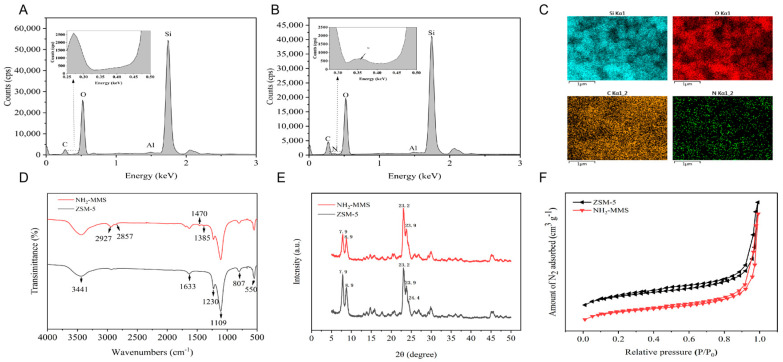
The element distribution of ZSM–5 (**A**) and NH_2_–MMS by EDS (**B**); the EDS mapping images for Si, O, C, and N elements of NH_2_–MMS (**C**); the FTIR of ZSM–5 and NH_2_–MMS (**D**); the XRD of ZSM–5 and NH_2_–MMS (**E**); the N_2_ adsorption–desorption isotherm and desorption isotherms of ZSM–5 and NH_2_–MMS (**F**).

**Figure 3 foods-13-02736-f003:**
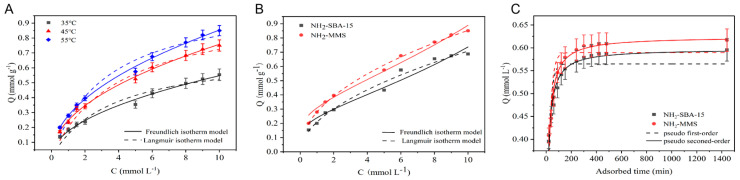
The static adsorption isotherm of NH_2_–MMS toward palmitic acid (C16:0) (**A**); the adsorption isotherms of NH_2_–MMS and NH_2_–SBA–15 toward palmitic acid (C16:0) at 55 °C (**B**); the dynamic adsorption isotherms of NH_2_–MMS and NH_2_–SBA–15 (**C**).

**Figure 4 foods-13-02736-f004:**
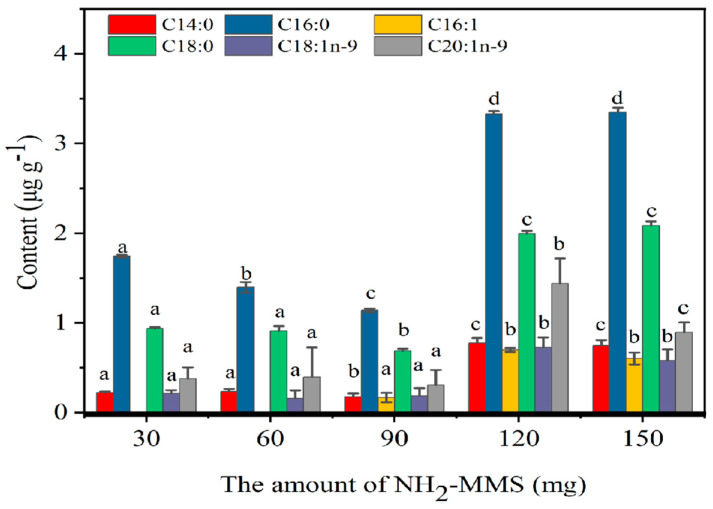
The effect of NH_2_-–MMS amount. Note: Different letters indicate that there are significant differences in fatty acids of different krill oil samples (*p* < 0.05).

**Figure 5 foods-13-02736-f005:**
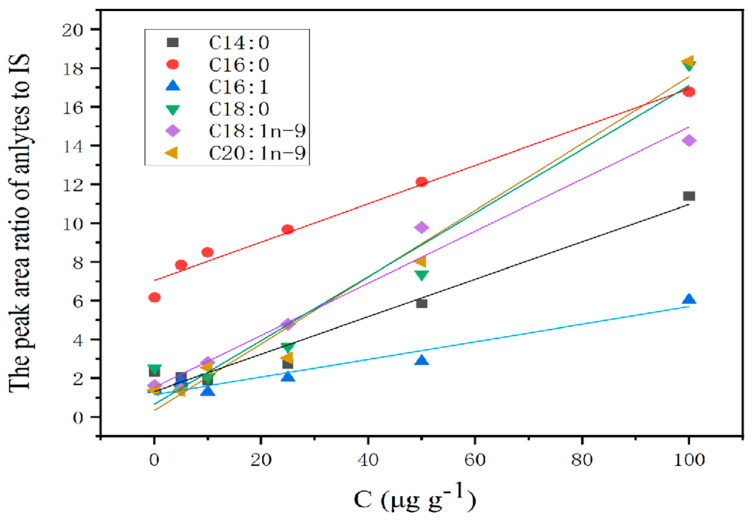
Standard curve for each FFA content.

**Table 1 foods-13-02736-t001:** Elemental composition of EDS in ZSM-5 and NH_2_-MMS.

Element	ZSM-5	NH_2_-MMS
C	16.69	31.59
O	44.53	35.84
Al	0.30	0.24
Si	38.48	31.82
N	0	0.50

**Table 2 foods-13-02736-t002:** The BET surface area, pore volume, and pore size of ZSM-5 and NH_2_-MMS.

	ZSM-5	NH_2_-MMS
BET surface area (m^2^/g)	220.04	163.58
Pore volume (cm^2^/g)	0.21	0.20
Pore size (nm)	10.31	12.93

**Table 3 foods-13-02736-t003:** Fitting parameters for Freundlich and Langmuir adsorption isotherm models of NH_2_-MMS and NH_2_-SBA-15 toward palmitic acid.

T (°C)	NH_2_-MMS						NH_2_-SBA-15					
	Langmuir Adsorption Isotherm Model			Freundlich Adsorption Isotherm Model			Langmuir Adsorption Isotherm Model			Freundlich Adsorption Isotherm Model		
	*q_m_*	*K_L_*	R^2^	*n*	*K_F_*	R^2^	*q_m_*	*K_L_*	R^2^	*n*	*K_F_*	R^2^
35	0.714	0.277	0.957	2.020	0.175	0.991	0.675	0.158	0.903	1.656	0.106	0.989
45	1.017	0.247	0.963	2.059	0.248	0.995	0.790	0.225	0.984	1.839	0.163	0.987
55	1.087	0.300	0.916	2.062	0.279	0.998	0.973	0.232	0.929	1.886	0.209	0.986

**Table 4 foods-13-02736-t004:** Related parameters for pseudo-first-order and pseudo-second-order adsorption isotherm of NH_2_-MMS and NH_2_-SBA-15 toward palmitic acid (C16:0) at 5 mmol L^−1^.

Adsorbents	Pseudo-First-Order Adsorption			Pseudo-Second-Order Adsorption		
	*q_e_*	*K* _1_	R^2^	*q_e_*	*K* _2_	R^2^
NH_2_-MMS	0.590	0.043	0.997	0.624	0.122	1.000
NH_2_-SBA-15	0.565	0.047	0.997	0.598	0.140	1.000

**Table 5 foods-13-02736-t005:** The linearity range, LODs, and LOQs of the current method (μg g^−1^).

Analytes	Linear Equation	R^2^	Linear Range	LODs	LOQs
C14:0	y = 6.203x + 1.463	0.998	0.10–100	0.06	0.20
C16:0	y = 5.805x + 7.404	0.996	0.10–100	0.06	0.20
C16:1	y = 1.464x + 1.432	0.992	0.25–100	0.15	0.50
C18:0	y = 6.612x + 1.624	0.998	0.10–100	0.06	0.20
C18:1n-9	y = 7.158x + 2.352	0.994	0.10–100	0.06	0.20
C20:1n-9	y = 7.549x + 1.497	0.999	0.25–100	0.15	0.50

**Table 6 foods-13-02736-t006:** The intraday and interday recoveries (%).

Analytes	Intraday Recoveries			Interday Recoveries		
	0.5 μg g^−1^	10 μg g^−1^	50 μg g^−1^	0.5 μg g^−1^	10 μg g^−1^	50 μg g^−1^
C14:0	87.88 ± 1.92	89.70 ± 0.08	91.12 ± 1.17	87.43 ± 1.68	88.20 ± 2.12	92.67 ± 3.40
C16:0	88.87 ± 0.86	91.07 ± 0.02	92.04 ± 0.19	88.88 ± 0.70	90.76 ± 0.48	93.35 ± 1.86
C16:1	90.85 ± 4.65	92.06 ± 4.18	94.10 ± 4.52	88.95 ± 4.65	89.36 ± 5.12	93.59 ± 3.76
C18:0	85.96 ± 2.75	88.72 ± 5.00	93.29 ± 1.10	86.97 ± 2.66	89.19 ± 4.13	92.55 ± 1.39
C18:1n-9	87.44 ± 3.70	89.59 ± 0.63	91.23 ± 1.91	86.89 ± 3.13	88.87 ± 1.14	90.55 ± 1.83
C20:1n-9	85.42 ± 2.55	88.12 ± 3.08	91.84 ± 2.80	85.16 ± 2.11	86.39 ± 3.51	94.31 ± 4.17

**Table 7 foods-13-02736-t007:** Fatty acid content of FFAs in Antarctic krill oil stored at −20 °C (μg g^−1^).

Analytes	0d	7d	14d	21d	28d	35d
C14:0	n.d.	n.d.	n.d.	n.d.	n.d.	0.68 ± 0.19 ^a^
C16:0	2.87 ± 0.18 ^d^	2.80 ± 0.12 ^d^	2.90 ± 0.26 ^c^	3.07 ± 0.81 ^b^	3.28 ± 0.61 ^ab^	3.42 ± 0.34 ^a^
C16:1n-7	n.d.	n.d.	n.d.	0.68 ± 0.18 ^bc^	0.80 ± 0.34 ^ab^	0.83 ± 0.51 ^a^
C18:0	1.00 ± 0.10 ^d^	0.97 ± 0.23 ^d^	1.11 ± 0.09 ^c^	1.22 ± 0.27 ^bc^	1.35 ± 0.19 ^b^	1.69 ± 0.19 ^a^
C18:1n-9	0.79 ± 0.40 ^b^	0.85 ± 0.72 ^a^	0.88 ± 0.37 ^a^	0.92 ± 0.33 ^a^	0.96 ± 0.26 ^a^	1.02 ± 0.01 ^a^
C20:1n-9	1.62 ± 0.24 ^c^	1.66 ± 0.21 ^c^	1.89 ± 0.09 ^bc^	1.93 ± 0.0.20 ^bc^	2.11 ± 0.11 ^b^	2.46 ± 0.36 ^a^

Note: Experimental data in the table are expressed as “mean ± standard deviation” (n = 3). Different letters marked after the same line indicate that there are significant differences in fatty acids of different krill oil samples (*p* < 0.05).

**Table 8 foods-13-02736-t008:** Fatty acid content of FFAs in Antarctic krill oil stored at 4 °C (μg g^−1^).

Analytes	0d	7d	14d	21d	28d	35d
C14:0	n.d.	n.d.	n.d.	0.71 ± 0.42 ^b^	0.92 ± 0.11 ^ab^	0.98 ± 0.19 ^a^
C16:0	2.70 ± 0.18 ^b^	2.79 ± 0.12 ^b^	2.98 ± 0.26 ^ab^	3.12 ± 0.81 ^ab^	3.28 ± 0.61 ^ab^	3.48 ± 0.34 ^a^
C16:1n-7	n.d.	n.d.	n.d.	n.d.	n.d.	0.78 ± 0.51 ^a^
C18:0	1.00 ± 0.10 ^c^	1.13 ± 0.23 ^bc^	1.23 ± 0.09 ^ab^	1.31 ± 0.27 ^ab^	1,44 ± 0.19 ^ab^	1.76 ± 0.19 ^a^
C18:1n-9	0.79 ± 0.40 ^e^	0.80 ± 0.72 ^e^	0.93 ± 0.37 ^d^	1.06 ± 0.33 ^c^	1.14 ± 0.26 ^b^	1.51 ± 0.01 ^a^
C20:1n-9	1.65 ± 0.24 ^e^	1.94 ± 0.21 ^d^	2.14 ± 0.09 ^c^	2.31 ± 0.20 ^bc^	2.44 ± 0.11 ^ab^	2.57 ± 0.36 ^a^

Note: Experimental data in the table are expressed as “mean ± standard deviation” (n = 3). Different letters marked after the same line indicate that there are significant differences in fatty acids of different krill oil samples (*p* < 0.05).

**Table 9 foods-13-02736-t009:** Fatty acid content of FFAs in Antarctic krill oil stored at 25 °C (μg g^−1^).

Analytes	0d	7d	14d	21d	28d	35d
C14:0	n.d.	n.d.	0.72 ± 0.12 ^cd^	0.92 ± 0.09 ^bc^	1.01 ± 0.34 ^b^	1.27 ± 0.48 ^a^
C16:0	2.70 ± 0.18 ^a^	2.73 ± 0.13 ^a^	2.87 ± 0.82 ^a^	3.20 ± 0.76 ^b^	3.28 ± 0.06 ^b^	3.45 ± 0.32 ^b^
C16:1n-7	n.d.	n.d.	n.d.	n.d.	0.76 ± 0.75 ^ab^	1.00 ± 0.09 ^a^
C18:0	1.00 ± 0.10 ^b^	1.12 ± 0.12 ^b^	1.16 ± 0.37 ^b^	1.21 ± 0.32 ^ab^	1.38 ± 0.15 ^ab^	1.56 ± 0.32 ^a^
C18:1n-9	0.79 ± 0.40 ^e^	1.10 ± 0.28 ^de^	1.31 ± 0.34 ^cd^	1.46 ± 0.24 ^bc^	1.79 ± 0.03 ^b^	2.38 ± 0.21 ^a^
C20:1n-9	1.65 ± 0.24 ^e^	2.09 ± 0.22 ^de^	2.21 ± 0.52 ^cd^	2.45 ± 0.29 ^bc^	2.65 ± 0.20 ^ab^	2.88 ± 0.29 ^a^

Note: Experimental data in the table are expressed as “mean ± standard deviation” (n = 3). Different letters marked after the same line indicate that there are significant differences in fatty acids of different krill oil samples (*p* < 0.05).

**Table 10 foods-13-02736-t010:** Fatty acid content of FFAs in Antarctic krill oil stored at 40 °C (μg g^−1^).

Analytes	0d	7d	14d	21d	28d	35d
C14:0	n.d.	0.68 ± 0.21 ^a^	0.89 ± 0.20 ^d^	1.11 ± 0.70 ^c^	1.35 ± 0.49 ^d^	1.58 ± 0.09 ^a^
C16:0	2.70 ± 0.18 ^e^	2.84 ± 0.30 ^a^	3.04 ± 0.56 ^e^	3.83 ± 0.4 ^c^	4.02 ± 0.06 ^d^	4.29 ± 0.19 ^b^
C16:1n-7	n.d.	n.d.	0.74 ± 0.28 ^e^	0.91 ± 0.3 ^c^	1.35 ± 0.14 ^d^	1.46 ± 0.25 ^b^
C18:0	1.00 ± 0.10 ^e^	1.31 ± 0.15 ^a^	1.65 ± 0.24 ^e^	2.11 ± 0.24 ^c^	2.50 ± 0.36 ^d^	2.76 ± 0.11 ^b^
C18:1n-9	0.79 ± 0.40 ^d^	1.01 ± 0.39 ^a^	1.28 ± 0.19 ^d^	1.50 ± 0.13 ^c^	2.31 ± 0.09 ^c^	2.87 ± 0.40 ^b^
C20:1n-9	1.65 ± 0.24 ^d^	2.08 ± 0.45 ^a^	2.58 ± 0.38 ^c^	2.83 ± 0.15 ^b^	3.67 ± 0.29 ^b^	4.18 ± 0.25 ^a^

Note: Experimental data in the table are expressed as “mean ± standard deviation” (n = 3). Different letters marked after the same line indicate that there are significant differences in fatty acids of different krill oil samples (*p* < 0.05).

**Table 11 foods-13-02736-t011:** Comparison of the current method with the reported methods.

Adsorbent Materials	Morphology	Pore Structure	Reaction Conditions	Toxins in the Reactive Materials	Loading Capacity	Literatures
Magnetic single-crystal ferrite nanoparticles	Spherical	-	High temperature and pressure	-	-	Wei Fang et al. [42]
Strong base anion exchange resin	Spherical	-	Normal temperature and pressure	Toluene and divinyl benzene	0.3–0.8 mmol g^−1^ (166.7–256.4 mg g^−1^)	Mhadmhan et al. [43]
Cassava peel materials	Bulk	Porous structure	High temperature and high pressure	-	0.8 mmol g^−1^ (322 mg g^−1^)	Phetrungnapha et al. [44]
Mesoporous silica nanoparticles	Rodlike	Mesoporous structure	Normal temperature and pressure	-	0.528 mmol g^−1^	Ahn et al. [45]
Amino-modified SBA-15	Cord-like domains	Mesoporous structure	Normal temperature and pressure	-	0.75 mmol g^−1^	Yang et al. [26]
NH_2_-MMS	Spherical	Hierarchical pores	Normal temperature and pressure	-	1.02 mmol g^−1^	Our study

## Data Availability

The data presented in this study are available on request from the corresponding author. The data are not publicly available due to privacy.

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
