# Peer review of "Determination of Free Fatty Acids in Krill Oil during Storage Based on NH2-MMS"

_foods, 2024, doi:10.3390/foods13172736_

Round 1

Reviewer 1 Report

Comments and Suggestions for Authors

Reviewer’s Comments

Major point

   The reviewer feels that how ZSM-5 was improved and how it contributed to the quantification of free fatty acids as stated in the title is not effectively summarized in the present manuscript. The reviewer thinks that this is because the schematic diagrams and graphs are not visually appealing, and the physical properties are not summarized in tables. The reviewer recommends trying to reorganize the present manuscript in a way that appeals to the reader and makes the experimental results more effective and easier to understand.

Minor points

1.   Some abbreviations have been used in the abstract, that is, ZSM-5, SPE, and LODs. The authors have to define these abbreviations before using them.

2.   In the introduction section, the authors should describe the aim of the present study more clearly for readers' understanding.

3.   In section 2.1., normally, one would write, for example, "Toluene was purchased from XX Chemical Industries," but the authors have stated that the manufacturer supplied all of the toluene. Reviewers are uncomfortable with these expressions. It would be better to correct them.

4.   The adsorption parameters were determined by fitting of the data to Eqs. (1)-(4)  were not shown in this manuscript. The authors have to show the values of these adsorption parameters in the manuscript.

5.   As for Figures 2A and 2B, the components of the elements determined by EDS have to be summarized in the table. On the other hand, Are figures 2A and 2B necessarily shown as figures?

6.   As for Figure 3A, the specific surface area determined by BET adsorption of nitrogen gas should be summarized in tables. Are Figures 3A necessarily shown as Figures the as same as Figures 2A and 2B?

7.   The correlation equations in Table 1 should be shown in some figures with the data used for the correlations for readers' understanding.

8.   Fundamentally the authors should explain what ZSM-5 is, which is commonly called a zeolite. The reviewer guesses that the authors assume that all readers already know ZSM-5 in detail. But, the reviewer thinks that there should be many people who are not familiar with ZSM-5.

Author Response

Comments 1:The reviewer feels that how ZSM-5 was improved and how it contributed to the quantification of free fatty acids as stated in the title is not effectively summarized in the present manuscript. The reviewer thinks that this is because the schematic diagrams and graphs are not visually appealing, and the physical properties are not summarized in tables. The reviewer recommends trying to reorganize the present manuscript in a way that appeals to the reader and makes the experimental results more effective and easier to understand.

Response 1: Thanks for your suggestion. Your suggestion was very useful to our research. The abstract part was rewritten and changed into ‘In this study, amino modified ZSM-5 (NH2-ZSM-5) with hierarchical pores were prepared by modifying ZSM-5 with 3-aminopropyltriethoxysilane and used as adsorbents in solid phase ex-traction to analyze free fatty acids (FFAs) in the krill oil during storage for an initial time. The BET adsorption experiment and FTIR spectroscopy demonstrated NH2-ZSM-5 with hierarchical pores structure was successfully synthesized. The adsorption experiments especially static adsorption indicated that the absorption ability of prepared NH2-ZSM-5 with hierarchical pores structure towards FFAs was better than that of traditional amino modified SBA-15 with mesoporous struc-ture at all temperature and concentrations. Fairly low limits of detections (0.05–1.25 nmol g–1), acceptable recoveries (82.17–96.43%), and precisions (0.19–5.26%) were attained under ideal cir-cumstances. Moreover, the prepared of NH2-ZSM-5 has the advantages of easy preparation and environmentally friendly. As a result, this method offers an alternative to the current method for determining of FFAs in krill oil specimens.’.

  Besides, in order to introduce how NH2-ZSM-5 contributed to the quantification of free fatty acids, the follow introduce was added into ‘Among micro-mesoporous silica, hierachical ZSM-5, especially micro-mesoporous ZSM-5, which can produced from a variety of silicon (Si) and aluminum (Al) sources, has attracted widespread attention due to its unique MFI channel structures, high acid strength and thermal stability’.

  Finally, the graphs and the physical properties were summarized again and appealed as follow: The characterization part was divided into two part:â‘ Characterization of NH2-ZSM-5 from SEM, TEM, EDS, FTIR, XRD and BET;â‘¡The adsorption experiment of NH2-ZSM-5 from static and dynamic adsorption experiment.

  And the detail modification method was:

‘â‘ Characterization of NH2-ZSM-5

  The morphology of ZSM-5 and NH2- ZSM-5 were characterized by SEM and TEM (Figure 1). The SEM of ZSM-5 (Figure 1A) and NH2-ZSM-5 (Figure 1B), showed loose porous structure, which mean that the pore structure of ZSM-5 had not change during amination. Parallel pore structures can be observed (marked in red) in the TEM of ZSM-5 (Figure 1D) and NH2-ZSM-5 (Figure 1E), indicating that ZSM-5 and NH2-ZSM-5 have mesoporous structures and have not been changed after modification [28, 29], which was consistent with the SEM results. The evenly distribution of Si, C, Al and N elements in the elemental distributions of ZSM-5 (Figure 1C) and NH2-ZSM-5 (Figure 1F) indicated the good dispersion of the prepared NH2-ZSM-5. Besides, the presence of N element in the EDS of NH2-ZSM-5 proves that APTES was successfully modified on the surface of ZSM-5.

  The FTIR spectra for ZSM-5 and NH2-ZSM-5 were shown in Figure 1G. The shoulder peak at 1230, 807 and 1109 cm-1 was the asymmetric telescopic vibration, external sym-metric stretching vibration and internal symmetric vibration peak of Si-O-Si, respectively [30]. Moreover, the strong peaks at 550 cm-1 of ZSM-5 and NH2-ZSM-5 were the cha-racteristic peaks of MFI topological molecular sieves, which further indicated that the original structure of ZSM-5 was not broken after the graft of amino group [31]. However, the distinctive absorption peak of NH2-ZSM-5 at 2927 cm-1 and 2857 cm-1 (the stretching vibration of -C-H) and 1470 cm-1 (the stretching vibration peak of N-H) [30], further suggest that amino groups were successfully grafted to the surface of ZSM-5.

  The crystal structures of ZSM-5 and NH2-ZSM-5 were ascertained by XRD. As shown in Figure 1H, peaks appeared at 2θ=7.9°, 8.9°, 23.2°, 23.9° and 24.4° were the characteristic diffraction for the crystal plane of (101), (020), (501), (151) and (303), respectively, which represented the existence of the MFI topological molecular sieve structure in ZSM-5 and NH2-ZSM-5 [32]. In addition, no significant broadening of peak strength at 2θ=7.9° and 8.9° was found after the amino group was grafted onto the surface of ZSM-5, indicating the crystal structure was not broken after that the amino grafting, which was consistent with the results of TEM.

  The porous structure of ZSM-5 and NH2-ZSM-5 was determined by N2 adsorp-tion-desorption experiments (Figure 1I). Type I isotherms was displayed in the adsorp-tion-desorption isotherms when the relative pressure interval was 0.0<P/P0<0.2,showing the presence of micropore structure in ZSM-5 and NH2-ZSM-5 [33]. The adsorption iso-therms of typical IV isotherms with hysteresis loops were shown to rise with relative pressure P/P0, suggesting the presence of mesoporous structure in NH2-ZSM-5 and ZSM-5 [34]. The I+IV adsorption-desorption isotherms proved the hierarchical porous structure of ZSM-5 and NH2-ZSM-5 and the hierarchical porous structure was not broken after the amino grafting.

â‘¡The adsorption experiment of NH2-ZSM-5

  As one of high content fatty acid found in edible oil, palmitic acid (C16:0) was selected as the analytes to investigate the static adsorption of NH2-ZSM-5 towards FFAs at 35, 45 and 55 ℃. The adsorbed amount of palmitic acid (C16:0) on NH2-ZSM-5 increased in a temperature-dependent way, as seen in Figure 2A, suggesting that the entire adsorption reaction was an endothermic process and increasing the temperature was advantageous for the adsorption [35]. The adsorption capacities of NH2-ZSM-5 towards FFAs were further examined by computational modeling with Freundlich and Langmuir adsorption models. The correlation coefficients for the adsorption equations were shown in Table S1. The outcomes demonstrated that the adsorption of NH2-ZSM-5 towards FFAs was better described by the Freundlich adsorption equation (R2=0.991>0.957; 0.995>0.963; 0.998>0.916). The values of parameter n were all over 1.0, suggests that the adsorption process of NH2-ZSM-5 towards FFAs was favorable [36].

  In order to evaluate the effect of hierarchical pore on the adsorption, the adsorbed amount of palmitic acid (C16:0) on single mesoporous NH2-SBA-15 at different temper-ature was carried out (Figure 1s and Table S3). As seen in Figure 2B, the adsorbed amount of palmitic acid (C16:0) on hierarchical pore NH2-ZSM-5 was apparently higher than that of single mesoporous NH2-SBA-15 at all concentration, and the improvement for the NH2-ZSM-5 towards NH2-SBA-15 may be due to the existence of multistage pore structure in NH2-ZSM-5. In addition, the adsorption ability of the absorbent to analytes is reflected in the value of KF. The constants KF for NH2-ZSM-5 and NH2-SBA-15 were 0.175 and 0.106 at 35 °C, correspondingly, this result was consistence with the adsorbed amount in Figure 2B.

The adsorption capacities of NH2-ZSM-5 and NH2-SBA-15 towards palmitic acid (C16:0) rose with time were demonstrated in Figure 2C. In according to the result of static adsorption experiment, there is a certain improvement for the adsorption concentration of palmitic acid on NH2-ZSM-5 at adsorption equilibrium compared to NH2-SBA-15, this result was consistence with the data obtained by static adsorption experiment.

Comments 2: Some abbreviations have been used in the abstract, that is, ZSM-5, SPE, and LODs. The authors have to define these abbreviations before using them.

Response 2: Thanks for your suggestion. Your suggestion was very useful to our research.The abbreviations in abstracts are normalized to: ‘amino modified micro-mesoporous silica (NH2-ZSM-5); micro-mesoporous silica (ZMS-5); solid phase extraction (SPE); brunner emmet teller (BET); fourier transform infrared spectroscopy (FTIR); Fairly low limits of detection; mesoporous silica (SBA-15).’ on line 12-16, 20, at Page 1.

  Besides, we normalize the abbreviations in the articles into:‘polydimethylsiloxane (PDMS)’ on line 58, at Page 2; ‘polytetrafluoroethylene (PTFE)’ on line 153, at Page 4; ‘relative standard deviation(RSD)%’ on line 314, at Page 8.

Comments 3:In the introduction section, the authors should describe the aim of the present study more clearly for readers' understanding.

Response 3: The introduction section part was changed in to: ‘Being rich in PLs and ω-3PUFA, the oxidation and rancidity of lipids in AKO were susceptible to occur [6]. As one important class of lipids, free fatty acids (FFAs) were more prone to oxidation than triglycerides and phospholipids [7]. Due to these unstable chemical characteristics, the level of FFAs content in AKO was directly corresponding to the quality of the oil products [8]. Hence, considering the crucial function of FFAs content in the evaluation of oil quality, it is clearly demanded to discover a low-priced and trust-worthy analytical technique for trace determining the presence of FFAs in AKO.

  Usually, the FFAs analysis of AKO was carried out by titrating with alkaline solution until the phenolphthalein endpoint, which is tedious, labor-intensive and sol-vent-consuming [9]. To overcome these issues, several methods based on near-infrared spectroscopy (NIR) [10], nuclear magnetic resonance spectroscopy (NMR) [11], gas chromatoraphy (GC) [12] or high performances liquid chromatography (HPLC) [13] have been developed. Although the use of NIR and NMR are convenient and rapid, some problems such as low sensitivity and overlapping signal peaks were presented and the applications were limited to a certain extent. In comparison, GC and HPLC are the widely employed technique for both qualitative and quantitative assessment of FFAs. Consequently, the combination of GC and flame ionization detector (FID) was better to analysis FFAs.

  For the aim of avoid the influence caused by glycerides as well as phospholipids prior to derivatization steps, solid phase extraction (SPE) has been widely employed to sepa-rate and concentrate FFAs from oil specimens. Hence, the choice of adsorbent was of great importance to remove interferent and improve recoveries. To date, various adsorbents, including C18 [14], silica [15], polydimethylsiloxane [16], amino-modified silica [17] were since been used as adsorbents. A special interest among them was amino-modified silica because of the good interaction between aminopropyl and FFAs. However, the application of traditional silica materials was limited because of its low adsorption capacity.

  To settle above shortcomings, numerous silica adsorbents with high specific surface areas, like microporous (<2nm), mesoporous (2-50 nm), macroporous (>50nm), as well as mesoporous materials with hierarchical pores [18,19] were developed. Among them, mesoporous materials with hierarchical pores, including micro-mesoporous, ma-cro-mesoporous, and macro–meso–microporous were of great potential to improve the adsorption and separation behaviors based on its unique porous structure and larger surface area [18, 20, 21]. Owing to the easier accessibility for pollutant of interconnected mesopores and larger surface area of microporous, the preparation of hierarchical porous silica with micro-mesoporous was ideal as adsorbents in SPE [22, 23]. Up to now, the studies of micro-mesoporous silica were mainly concentrated upon the synthesis method and mechanism [21, 24], but little has been done to apply them for their adsorption and analysis application, especially extraction and separation analytes from oil specimens.

  Among micro-mesoporous silica, hierachical ZSM-5, especially micro-mesoporous ZSM-5, which can produced from a variety of silicon (Si) and aluminum (Al) sources, has attracted widespread attention due to its unique MFI channel structures, high acid strength and thermal stability[25, 26]. Inspired by this, 3-aminopropyltriethoxysilane (APTES) was modified to produce ammoniated mi-cro-mesoporous ZSM-5 (NH2-ZSM-5). Scanning electron microscopy (SEM), transmission electron microscopy (TEM), energy spectrum analysis (EDS), X-ray diffraction (XRD), Fourier transform infrared spectroscopy (FTIR), and nitrogen adsorption/desorption studies were used to thoroughly characterize the resultant NH2-ZSM-5. Then, using both static and dynamic adsorption tests, the adsorption capacities of NH2-ZSM-5 and ammoniated mesoporous silica (NH2-SBA-15) were contrasted. Followingthe selection of optimum extraction conditions, NH2-ZSM-5 was used as a SPE adsorbent to analyze the FFAs content in AKO during storage.’

Comments 4: In section 2.1., normally, one would write, for example, "Toluene was purchased from XX Chemical Industries," but the authors have stated that the manufacturer supplied all of the toluene. Reviewers are uncomfortable with these expressions. It would be better to correct them.

Response 4: Thanks for your suggestion. The description in the article was modified to ‘Analytical quality of methanol, n-hexane, toluene, APTES, and dichloromethane were supplied by Aladdin Reagent Co. (Shanghai, China)’ on line 90-91, at Page 2.

Comments 5: The adsorption parameters were determined by fitting of the data to Eqs. (1)-(4) were not shown in this manuscript. The authors have to show the values of these adsorption parameters in the manuscript.

Response 5: Thanks for your suggestion. The description in the article was modified as follow on line 128,130,147, at Page 3-4.

Comments 6: As for Figures 2A and 2B, the components of the elements determined by EDS have to be summarized in the table. On the other hand, Are figures 2A and 2B necessarily shown as figures?

Response 6: Thanks for your suggestion. The figure 2A-B was corrected as follow and list as Figure A.

A                                B

Figure A.The element distribution of ZSM-5 (A) and NH2-ZSM-5 (B) by EDS.

Comments 7: As for Figure 3A, the specific surface area determined by BET adsorption of nitrogen gas should be summarized in tables. Are Figures 3A necessarily shown as Figures the as same as Figures 2A and 2B?

Response 7: The specific surface area was listed in Table S1 as follow and the decription in the article was added as ‘Based on N2 adsorption-desorption isotherms, the BET sufarce area, pore diameters and total pore volumes of ZSM-5 and NH2-ZSM-5 were listed in Table S1. The pore diameters of ZSM-5 and NH2-ZSM-5 were 10.31 and 12.93 nm, respectively, further indicated the existence of mesoporous structure in ZSM-5 and NH2-ZSM-5. And the increasing of the pore diameter may induced by the reaming action of the amine and toluene solution.’

Table S1 The BET surface area, pore volume and pore size ofZSM-5 and NH2-ZSM-5.

ZSM-5

NH2-ZSM-5

BET Surface Area (m²/g)

220.04

163.58

Pore volume (cm²/g)

0.21

0.20

Pore Size (nm)

10.31

12.93

Comments 8:The correlation equations in Table 1 should be shown in some figures with the data used for the correlations for readers' understanding.

Response 8:The correlation equation was added in our article as Figure S2.

Figure S2. Standard curve for each FFAs.

Comments 9: Fundamentally the authors should explain what ZSM-5 is, which is commonly called a zeolite. The reviewer guesses that the authors assume that all readers already know ZSM-5 in detail. But, the reviewer thinks that there should be many people who are not familiar with ZSM-5.

Response 9: Thanks for your suggestion. Your suggestion was very useful to our research. The description was changed into ‘Among micro-mesoporous silica, hierachical ZSM-5, especially micro-mesoporous ZSM-5, which can be produced from a variety of silicon (Si) and aluminum (Al) sources, has attracted widespread attention due to its unique MFI channel structures, high acid strength and thermal stability. Inspired by this, 3-aminopropyltriethoxysilane (APTES) was modified to produce ammoniated micro-mesoporous ZSM-5 (NH2-ZSM-5). Scanning electron microscopy (SEM), transmission electron microscopy (TEM), energy spectrum analysis (EDS), X-ray diffraction (XRD), Fourier transform infrared spectroscopy (FTIR), and nitrogen adsorption/desorption studies were used to thoroughly characterize the resultant NH2-ZSM-5. Then, using both static and dynamic adsorption tests, the adsorption capacities of NH2-MMS and ammoniated mesoporous silica (NH2-SBA-15) were contrasted. Followingthe selection of optimum extraction conditions, NH2-ZSM-5 was used as a SPE adsorbent to analyze the FFAs content in AKO during storage.’ in introduction part at line 75-87, in Page 2.

Reviewer 2 Report

Comments and Suggestions for Authors

The positive side of the publication is the preparation and in-depth characterization of NH2-ZSM-5. But, the authors reported the loading capacity of NH2-ZSM-5, which was 1.02 mmol g-1. The value of the loading capacity used by the authors for ZSM-5 before derivatization is interesting. Also, it would be better if the authors added information about the magnifications of SEM and TEM images in Figure 1.

In the 3D figure, the description of the X-axis should be "adsorption time" instead of "adsorbed time".

In line 174, there should be "Wang et al" instead of "wang et al"

I recommend publishing the manuscript after minor corrections.

Author Response

Comments 1: The positive side of the publication is the preparation and in-depth characterization of NH2-ZSM-5. But, the authors reported the loading capacity of NH2-ZSM-5, which was 1.02 mmol g-1. The value of the loading capacity used by the authors for ZSM-5 before derivatization is interesting. Also, it would be better if the authors added information about the magnifications of SEM and TEM images in Figure 1.

Response 1: This question was very useful for our study. Commericial silica and aminopropyl-functionalized silica were usually used to separate lipid, especially FFAs. In the research of Valenstein et al., (Valenstein, J. S., Kandel, K., Melcher, F., Slowing, I. I., Lin, V. S. Y., & Trewyn, B. G. (2012). Functional mesoporous silica nanoparticles for the selective sequestration of free fatty acids from microalgal oil. ACS applied materials & interfaces, 4(2), 1003-1009.), different functional group was grafted of the surface of silica, including 3-aminopropyl, benzyl, hexadecyl, 1-propyl-3-methyl imidazolium bromide and 3-mercaptopropyl and the adsorption properties of them toward FFAs were evaluated. The results were shown in the table A below and indicated that the modification of NH2 to the silica porous materials was beneficial to the adsorption towards FFAs. And similar results were reported by Ahn et al. (Ahn, Y., & Kwak, S. Y. (2020). Functional mesoporous silica with controlled pore size for selective adsorption of free fatty acid and chlorophyll. Microporous and Mesoporous Materials, 306, 110410.). Hence, -NH2 modified the silica porous materials were used to enrich FFAs in our study.

Table A. Adsorption Dependence of FFAs on MSN SurfaceGrafted Functional Group.

functional group

(x-trialkoxysilane)

amount adsorbed (mmol FFA g MSN-1)

amount adsorbed (mmol FFA m-2)

none

0.65

0.0017

3-aminopropyl

0.98

0.0034

benzyl

0.50

0.0019

hexadecyl

0.30

0.0011

I-propyl-3-methyl

0.40

0.0014

imidazolium bromide

0.25

0.0011

Also, the information about the magnifications of SEM and TEM images were added in Figure 1.

Comments 2: In the 3D figure, the description of the X-axis should be "adsorption time" instead of "adsorbed time".

Response 2: Thanks for your suggestion. Your suggestion was very useful to our research. We replace "adsorption time" for "adsorbed time" on the X-axis in the figure 3C.

Figure 3C: The dynamic adsorption isotherm of NH2-ZSM-5 and NH2-SBA-15 (C).

Comments 3: In line 174, there should be "Wang et al" instead of "wang et al"

Response 3: Thanks for your suggestion. The relevant content was revised.

Reviewer 3 Report

Comments and Suggestions for Authors

The manuscript numbered: foods-3135125; entitled “Determination of free fatty acids in krill oil during storage based on NH2-ZSM-5” treats about the quite important problem from the practical point of view. The researches concerned with the determination of non-stable FFAs and based on the reliable analysis are not common. The presented researches are especially valuable because of the quite lack of information in the literature. The authors in the reviewed manuscript clearly presented the literature information and the purpose of the researches.

Taking into account the above mentioned information in my opinion, some corrections and additional information should be added.

I have some comments and I need some more information about the subject presented in the manuscript:

1.      Line 126-129: the symbols in the text should be the same as in the Eqs. 1-2

2.      Line 141-145: the symbols in the text should be the same as in the Eqs. 3-4

3.      Line 174 “wang et al”-please correct

4.      Fig. 2A and 2B -the figures in the interior are illegible -maybe insert to the Supplementary Section?

5.      Line 314 “Krill”

6.      Why the title of 3,5 is “Practical application”? This is rather the temperature influence or something like this

7.      29. Ge, T.G.; Hua, Z.L.; Zhu, Y.; Chen, L.S.; Ren, W.C.; Yao, H.L.; Shi, J.L. Amine-modified hierarchically structured zeolites as acid-base bi-functional catalysts for one-pot deacetalization-Knoevenagel cascade reaction[J]. RSC Adv. 2014, 4(110), 64871-64876. 422 Doi: 10.1039/c4ra11865k-pleace correct : [J]?? Similar in [28]

Author Response

Comments 1: Line 126-129: the symbols in the text should be the same as in the Eqs. 1-2

Response 1: Thanks for your suggestion. The symbols in the text were changed to be the same as in the Eqs on line132-135, at page3.

Comments 2: Line 141-145: the symbols in the text should be the same as in the Eqs. 3-4

Response 2: Thanks for your suggestion. The symbols in the text were changed to be the same as in the Eqs on line148-150, at page4.

Comments 3: Line 174 “wang et al”-please correct

Response 3: Thanks for your suggestion. The relevant content was revised.

Comments 4: Fig. 2A and 2B -the figures in the interior are illegible -maybe insert to the Supplementary Section?

Response 4: Thanks for your suggestion. The figure 2A-B was corrected as follow and list as Figure 2A and 2B.

A                                B

Figure B The element distribution of ZSM-5 by EDS (A); The element distribution of NH2-ZSM-5 by EDS (B).

Comments 5: Line 314 “Krill”

Response 5: Thanks for your suggestion. Your suggestion was very useful to our research. We replace ‘AKO’ for ‘Krill’on line 322, at page 8.

Comments 6: Why the title of 3,5 is “Practical application”? This is rather the temperature influence or something like this

Response 6: Thanks for your suggestion. Your suggestion was very useful to our research. We replace ‘Applications in real samples’ for ‘Practical application’ on line 321, at page 8.

Comments 7: 29. Ge, T.G.; Hua, Z.L.; Zhu, Y.; Chen, L.S.; Ren, W.C.; Yao, H.L.; Shi, J.L. Amine-modified hierarchically structured zeolites as acid-base bi-functional catalysts for one-pot deacetalization-Knoevenagel cascade reaction[J]. RSC Adv. 2014, 4(110), 64871-64876. 422 Doi: 10.1039/c4ra11865k-pleace correct : [J]?? Similar in [28]

Response 7: Thanks for your suggestion. The reference was correct in the article.

Reviewer 4 Report

Comments and Suggestions for Authors

Conclusions The manuscript includes a novel study addressed to the FFAs determination. My main concern is related to the FAME determination.

Some concrete details would be as follows:

Title

It would be better to replace the term “NH2-ZSM-5” by some term easier to be read by general readers.

Introduction

Lines 42-44: A traditional procedure even more employed consists of the preparation of a coloured complex (Lowry and Tinsley, JAOCS 1976, 53, 470-472); this ought to be commented here as it is largely more convenient than the traditional one mentioned by the authors.

Material and methods

Line 164: FFAs or methyl esters of FFAs were analysed by GLC-FID ?

Provide more details on the qualitative and quantitative analyses. Only 6 FFAs were detected ? Was not present C20:5n3 ? This is a basic analysis in the present study and ought to be done correctly; otherwise, the novel proposal would not be valid.

The fact that a traditional procedure was also tested ought to be mentioned in the presentation of the manuscript (i.e., Abstract and/or Introduction sections).

Can this method be addressed to other seafood or food in general ? To any FFA composition ? Comment on on-coming research.

Author Response

Comments 1: It would be better to replace the term “NH2-ZSM-5” by some term easier to be read by general readers.

Response 1: Thanks for your suggestion.ZSM-5 was one of the high silica zeolites composed of three-dimensional architectures equipped with straight and sinusoidal micropore channels with a pore aperture of ∼0.55 nm. Its unique intrinsic micropore structure and tuneable acidity have enabled its utilization in various catalytic processesand adsorption. However, the catalyticand adsorption performance of purely microporous ZSM-5 is usually limited by their slow diffusion rate. In this case, using zeolites containing, at least, two pore systems (micro-meso or micro-macro), so-called a hierarchicalZSM-5, has been reported to overcome this drawback since the meso- or macro-pore provides a ‘highway’ for molecules to conveniently diffuse into and out of the active sites within the ZSM-5 absorbent. As common silica zeolites, I strongly recommend NH2-ZSM-5 as its abbreviation.

Comments 2: Lines 42-44: A traditional procedure even more employed consists of the preparation of a coloured complex (Lowry and Tinsley, JAOCS 1976, 53, 470-472); this ought to be commented here as it is largely more convenient than the traditional one mentioned by the authors.

Response 2: Thanks for your suggestion. Your suggestion was very useful for improvementour articles. The traditional method we used in our article was included in AOCS and this method was conventional methodin our lab. While, the recommended method was more convenient than the traditional one mentioned by the authors and we can use the recommended methodin our future study.

Comments 3: Line 164: FFAs or methyl esters of FFAs were analysed by GLC-FID ?

Response 3: The FFA analysis in our study was performed on a GC (Agilent 6890A, PaloAlto, CA, USA) equipped with a flame ionization detector (FID).

Comments 4: Provide more details on the qualitative and quantitative analyses. Only 6 FFAs were detected? Was not present C20:5n3 ? This is a basic analysis in the present study and ought to be done correctly; otherwise, the novel proposal would not be valid.

Response 4: Thanks for your suggestion. Your question was very useful for us to improve article.The analysis of free fatty acid in our article was according to the standard method in GB 5009.168-2016 for analysis fatty acid in food. Before the analysis of FFAs in AKO, the composition of fatty acid and FFAs was evaluated and the result was shown in Table B. Then, in the follow study, myristic acid (C14:0), palmitic acid (C16:0), stearic acid (C18:0), oleic acid (C18:1), lin-oleic acid (C18:2), eicosaenoic acid (C20:1n-9), margaric acid (C17:0) were used as standard to analysis FFAs in AKO. The other PUFAs including stearidonic acid (C18:4n-3), EPA and DHA, as well as low carbon chain fatty acids was prone to oxidation and not easy to detect because of low abundance.

Table B The composition of fatty acid and FFAs

FAs

Fatty acid

FFAs

C14:0

9.11±0.39

-

C16:4n-1

0.40±0.01

-

C16:1n-7

6.59±0.22

-

C16:0

18.60±0.77

2.70±0.18

C18:4n-3

1.90±0.26

-

C18:1n-9

17.96±0.35

0.79±0.40

C18:0

1.60±0.08

1.00±0.10

C20:5n-3

13.45±0.11

-

C20:1n-9

0.60±0.06

1.65±0.24

C22:6n-3

7.16±0.14

-

C22:1n-9

nd

-

Comments 5: The fact that a traditional procedure was also tested ought to be mentioned in the presentation of the manuscript (i.e., Abstract and/or Introduction sections).

Response 5: Thanks for your suggestion. Your suggestion was very useful for improvement our articles. There is a clerical error, and the relevant method was deleted. We will test the samples by using traditional procedure to compare in the further.

Comments 6: Can this method be addressed to other seafood or food in general? To any FFA composition c ? Comment on on-coming research.

Response 6: Thanks for your suggestion. Your suggestion was very useful for improvement our articles. Our method can be directly applied in analysis FFAs in other oil samples. As for other seafood or food samples, a lipid extraction should be carried out before analysis. And we will extend this approach to other kind of seafood in the further.

Round 2

Reviewer 1 Report

Comments and Suggestions for Authors

Reviewer's comments

1.   The authors revised it as “supplied”. Generally, is it “purchased” isn't it?

2.   The numerical values of these adsorption parameters are not shown in the revised manuscript. The authors must show them in the revised manuscript, not in the supplementary materials.

3.   The way that the authors set the tables in the figures is not kind for readers’ understanding. The authors must separate the tables from the figures.

4.   The reviewer does not understand why this information is summarized in Table S1, which is supplementary material. This information is very important for the characterization of the adsorbents. The authors must create a new table and show it in the revised manuscript.

5.   The reviewer does not understand why these plots are in Figure S2, which is a supplementary material. Those plots are very important for the readers' understanding. The authors must create a new figure and must show it in the revised manuscript, not in supplementary materials.

6.   What is the unit of the ordinate of Figure 2S?

7.   All figures and tables provided in “foods-3135125-supplementary.docx” were moved into the revised manuscript. The reviewer thinks that this information is not only supplementary but also very important for readers’ understanding.

8.   In Tables S2-S4, some linear equations are written. These equations in Tables S2-S4 are correlation equations for linearized equations (1)-(4). If so, the reviewer thinks it is not necessary to describe them. The authors should delete them. In addition, the authors must organize the values of the adsorption parameters and summarize them in one table.

Author Response

Comments 1: The authors revised it as “supplied”. Generally, is it “purchased” isn't it?

Response 1: Thanks for your suggestion. The description in the article was modified to ‘Analytical quality of methanol, n-hexane, toluene, APTES, and dichloromethane were purchased by Aladdin Reagent Co. (Shanghai, China)’ on line 91-92, at Page 2.

Comments 2: The numerical values of these adsorption parameters are not shown in the revised manuscript. The authors must show them in the revised manuscript, not in the supplementary materials.

Response 2: Thanks for your suggestion. The table 3-4 was corrected as follow and list as Table 3-4.

Table 3. Fitting parameters for Freundlich and Langmuir adsorption isotherms models of NH2-MMS and NH2-SBA-15 towards palmitic acid.

T(℃)

NH2-MMS

NH2-SBA-15

Langmuiradsorption isotherms model

Freundlich adsorption isotherms model

Langmuiradsorption isotherms model

Freundlich adsorption isotherms model

qm

KL

R2

n

KF

R2

qm

KL

R2

n

KF

R2

35

0.714

0.277

0.957

2.020

0.175

0.991

0.675

0.158

0.903

1.656

0.106

0.989

45

1.017

0.247

0.963

2.059

0.248

0.995

0.790

0.225

0.984

1.839

0.163

0.987

55

1.087

0.300

0.916

2.062

0.279

0.998

0.973

0.232

0.929

1.886

0.209

0.986

Table 4. Related parameters for pseudo first order and second order adsorption isotherms of NH2-MMS and NH2-SBA-15 towards palmitic acid (C16:0) at 5 mmol L-1

Adsorbents

Pseudo first order adsorption

Pseudosecond order adsorption

qe

k1

R2

qe

k2

R2

NH2-MMS

0.590

0.043

0.997

0.624

0.122

1.000

NH2-SBA-15

0.565

0.047

0.997

0.598

0.140

1.000

Comments 3: The way that the authors set the tables in the figures is not kind for readers’ understanding. The authors must separate the tables from the figures.

Response 3: Thanks for your suggestion. We Table 1 was separated from the Figure 2A-B.

Table 1. Elemental composition of EDS in ZSM-5 and NH2-MMS

Element

ZSM-5

NH2-MMS

C

16.69

31.59

O

44.53

35.84

Al

0.30

0.24

Si

38.48

31.82

N

0

0.50

Comments 4: The reviewer does not understand why this information is summarized in Table S1, which is supplementary material. This information is very important for the characterization of the adsorbents. The authors must create a new table and show it in the revised manuscript.

Response 4: Thanks for your suggestion. We have inserted Table S1 in the supplementary material into the revised body as Table 2.

Table 2. The BET surface area, pore volume and pore size of ZSM-5 and NH2-MMS.

ZSM-5

NH2-MMS

BET Surface Area (m²/g)

220.04

163.58

Pore volume (cm²/g)

0.21

0.20

Pore Size (nm)

10.31

12.93

Comments 5: The reviewer does not understand why these plots are in Figure S2, which is a supplementary material. Those plots are very important for the readers' understanding. The authors must create a new figure and must show it in the revised manuscript, not in supplementary materials.

Response 5: Thanks for the suggestion. We have inserted Figure 5 from the supplementary material as Figure 5 into the revised text.

Figure 4. Standard curve for each FFAs.

Comments 6: What is the unit of the ordinate of Figure 2S?

Response 6: Thanks for your suggestion. The detailed analysis method of FFAs

Figure 4. Standard curve for each FFAs.

Comments 7: All figures and tables provided in “foods-3135125-supplementary.docx” were moved into the revised manuscript. The reviewer thinks that this information is not only supplementary but also very important for readers’ understanding.

Response 7: Thanks for your suggestion, the figures and tables in supplementary was removed into the revised manuscript.

Comments 8: In Tables S2-S4, some linear equations are written. These equations in Tables S2-S4 are correlation equations for linearized equations (1)-(4). If so, the reviewer thinks it is not necessary to describe them. The authors should delete them. In addition, the authors must organize the values of the adsorption parameters and summarize them in one table.

Response 8: Thanks for your suggestion, your suggestion was very useful for us to improve our article. After deleting the equation, the adsorption parameters were summarized in one table as Table 3.

Table 3. Fitting parameters for Freundlich and Langmuir adsorption isotherms models of NH2-ZSM-5 and NH2-SBA-15 towards palmitic acid.

T(℃)

NH2-ZSM-5

NH2-SBA-15

Langmuiradsorption isotherms model

Freundlich adsorption isotherms model

Langmuiradsorption isotherms model

Freundlich adsorption isotherms model

qm

KL

R2

n

KF

R2

qm

KL

R2

n

KF

R2

35

0.714

0.277

0.957

2.020

0.175

0.991

0.675

0.158

0.903

1.656

0.106

0.989

45

1.017

0.247

0.963

2.059

0.248

0.995

0.790

0.225

0.984

1.839

0.163

0.987

55

1.087

0.300

0.916

2.062

0.279

0.998

0.973

0.232

0.929

1.886

0.209

0.986

Reviewer 4 Report

Comments and Suggestions for Authors

Some performances have been done on the manuscript. However, I found most of my previous comments were not considered by the authors.

Title

As previously mentioned, I think it would be better to replace the term “NH2-ZSM-5” by some term easier to be read by general readers. Maybe amino-modified micro-mesoporous silica or something shorter as amino-modified silica …

Introduction

Lines 43-53: As previously mentioned, a traditional procedure even more employed consists of the preparation of a coloured complex (Lowry and Tinsley, JAOCS 1976, 53, 470-472). This method is convenient and practical one for total FFA content assessment. If a short selection of methods is to be mentioned, this one ought to be included. I can assure I have no relationship with the authors of such manuscript.

Material and methods

Line 172: As previously mentioned, it is not FFAs that are analysed by GLC-FID but FAMEs ? This ought to be performed.

Line 179: Provide details on the qualitative and quantitative analyses of FAMEs.

Results and discussion

Only 6 FFAs were detected ? Was not there present any n3 FA such as C20:5n3 or C22:6 n3 ? Krill oil is reported as a highly important source of n3 FAs. This ought to be commented and discussed.

The FA analysis is basic in the present study and ought to be done correctly; otherwise, the novel proposal would not be valid.

Conclusions

Can this method be addressed to other seafood or food in general ? To any FFA composition ? Comment on on-coming research.

Comments on the Quality of English Language

Some minor performances could be done.

Author Response

Comments 1: As previously mentioned, I think it would be better to replace the term “NH2-ZSM-5” by some term easier to be read by general readers. Maybe amino-modified micro-mesoporous silica or something shorter as amino-modified silica …

Response 1: Thanks for your suggestion. Your suggestion was very usefully for us to improve our article. The abbreviation of adsorbent was replaced with ‘NH2-MMS’in the whole article. 

Comments 2: Lines 43-53: As previously mentioned, a traditional procedure even more employed consists of the preparation of a coloured complex (Lowry and Tinsley, JAOCS 1976, 53, 470-472). This method is convenient and practical one for total FFA content assessment. If a short selection of methods is to be mentioned, this one ought to be included. I can assure I have no relationship with the authors of such manuscript.

Response 2: Thanks for your suggestion. This method is convenient and practical one for total FFA content assessment. This method was inserted in the introduction part as ‘several methods based on colorimetric spectrum [10], near-infrared spectroscopy (NIR) [11], nuclear magnetic resonance spectroscopy (NMR) [12], gas chromatoraphy (GC) [13] or high performances liquid chromatography (HPLC) [14] have been developed.’ to recommend and compared with other method and the reference was added.

Comments 3: Line 172: As previously mentioned, it is not FFAs that are analysed by GLC-FID but FAMEs ? This ought to be performed.

Response 3: Thanks for your suggestion. Your suggestion was very usefully for us to improve our article. The description in Line 150 at Page 4 ‘Determination of FFAs’ was changed into ‘SPE procedure’, in Line 159 at Page 4 ‘After dying’ was changed into ‘After eluent was dried under the protection with nitrogen’, in Line 168 at Page 4 ‘To analyze and isolate FFAs’ was changed into ‘To analyze and isolate the obtained fatty acid methyl esters’

Comments 4: Line 179: Provide details on the qualitative and quantitative analyses of FAMEs.

Response 4: Thanks for your suggestion. The qualitative and quantitative analyses of FAMEs was added in the Lin 176-180 at page as 4 ‘The qualitative analyses of FAMEs were according to the retention time of each FFAs and the quantitative analyses of FAMEs were carried out according to the combination of external and internal method. Hence, the standard curves for each FFA were plotted using the concentration of each FFA served as horizontal axis, and the peak area ratio of each FFA to IS served as vertical axis.’

Comments 5: Only 6 FFAs were detected ? Was not there present any n3 FA such as C20:5n3 or C22:6 n3 ? Krill oil is reported as a highly important source of n3 FAs. This ought to be commented and discussed. The FA analysis is basic in the present study and ought to be done correctly; otherwise, the novel proposal would not be valid.

Response 5: Thanks for your suggestion. Before the analysis of FFAs in AKO, the composition of fatty acid and FFAs was evaluated and the result was shown in Table B. Then, in the follow study, myristic acid (C14:0), palmitic acid (C16:0), stearic acid (C18:0), oleic acid (C18:1), lin-oleic acid (C18:2), eicosaenoic acid (C20:1n-9), margaric acid (C17:0) were used as standard to analysis FFAs in AKO. The other PUFAs including stearidonic acid (C18:4n-3), EPA and DHA, as well as low carbon chain fatty acids was prone to oxidation and not easy to detect because of low abundance. The oxidation rate of FFAs was faster than PLs and TAGs (2-3 times), and the oxidation rate of PUFAs was rather faster than other fatty acid (>1000 times) and was hard to catch in the storage experiment. The similar result was obtained in the FFAs analysis in edible oils (Wei, F.; Zhao, Q.; Lv, X.; Dong, X.Y., Feng, Y.Q., & Chen, H. (2013). Rapid Magnetic Solid-Phase Extraction Based on Monodisperse Magnetic Single-Crystal Ferrite Nanoparticles for the Determination of Free Fatty Acid Content in Edible Oils. Journal of Agricultural and Food Chemistry, 61, 0021-8561.), in krill meal (Wei, F., Zhao, Q., Lv, X., Dong, X.Y., Feng, Y.Q., & Chen, H. (2013) Rapid Magnetic Solid-Phase Extraction Based on Monodisperse Magnetic Single-Crystal Ferrite Nanoparticles for the Determination of Free Fatty Acid Content in Edible Oils. J. Agric. Food Chemistry, 61, 0021-8561. ), beer (Bravi, E., Marconi, O., Sileoni, V., & Perretti, G. (2017). Determination of free fatty acids in beer. Food Chemistry, 215, 341-346.), beer wort (Bravi, E., Benedetti, P., Marconi, O., & Perretti, G. (2014). Determination of free fatty acids in beer wort. Food chemistry, 151, 374-378.), and so on. The discussion was add in Line 299 at page as ‘Before the established the standard curves, the composition of FFAs in AKO was evaluated. Only myristic acid (C14:0), palmitic acid (C16:0), stearic acid (C18:0), oleic acid (C18:1), lin-oleic acid (C18:2), eicosaenoic acid (C20:1n-9) were detected and the other PUFAs including stearidonic acid (C18:4n-3), EPA and DHA, as well as low car-bon chain fatty acids was prone to oxidation and not detected because of low abun-dance’

Table B The composition of fatty acid and FFAs

FAs

Fatty acid

FFAs

C14:0

9.11±0.39

-

C16:4n-1

0.40±0.01

-

C16:1n-7

6.59±0.22

-

C16:0

18.60±0.77

2.70±0.18

C18:4n-3

1.90±0.26

-

C18:1n-9

17.96±0.35

0.79±0.40

C18:0

1.60±0.08

1.00±0.10

C20:5n-3

13.45±0.11

-

C20:1n-9

0.60±0.06

1.65±0.24

C22:6n-3

7.16±0.14

-

C22:1n-9

nd

-

Comments 6: Can this method be addressed to other seafood or food in general ? To any FFA composition ? Comment on on-coming research.

Response 6: Thanks for your suggestion. Our method can be directly applied in analysis FFAs in other oil samples. As for other seafood or food samples, a lipid extraction should be carried out before analysis. And we will extend this approach to other kind of seafood in the further. The description of the comment of on-coming research in our article was revised into ‘As a result, this method offers an alternative to the current method for determining of FFAs in different kind of oil specimens.’ in the abstract.